# Paternal effects without paternity? Testing non-genetic male influence on offspring size and brood size in a gynogenetic vertebrate, the Amazon molly (*Poecilia formosa*)

Ulrike Scherer[1,2,3]*, Sean M. Ehlman[1,2,3,4], David Bierbach[1,2,3], Jens Krause[1,2,3], Max Wolf[1,3]

1 SCIol Excellence Cluster, Technische Universität Berlin, Berlin, Germany, 2 Faculty of Life Sciences, Humboldt University, Berlin, Germany, 3 Department of Fish Biology, Fisheries, and Aquaculture, Leibniz Institute of Freshwater Ecology and Inland Fisheries, Berlin, Germany, 4 Department of Biological Sciences, University of South Carolina, Columbia, South Carolina, United States of America

* u.k.scherer@gmail.com

## Abstract

Paternal effects, i.e., effects of fathers on the phenotype of their offspring that are not mediated by the transmission of alleles, are increasingly recognized as a potentially significant source of phenotypic variation across taxa – even in the absence of paternal care. Gynogenetic systems, which rely on sperm to trigger embryogenesis without incorporating male genetic material, provide a powerful way to experimentally isolate paternal effects from effects caused by the integration of male genetic material. Up to now, however, paternal effects remain largely unexplored in these systems. Here, we tested for non-genetic paternal effects in the gynogenetic Amazon molly (*Poecilia formosa*): a naturally clonal, all-female species with no parental care. Using a highly controlled breeding experiment involving 60 Atlantic molly males (*Poecilia mexicana*) and 54 Amazon molly females from a single clonal lineage, we generated 128 broods and 2,435 offspring. While males were drawn from a naturally variable stock population, females – next to being genetically identical – were standardized for age, size, descent, and developmental experience. We asked whether male identity or body size predicted offspring size – a key offspring phenotypic trait. We also asked whether male identity or body size predicted brood size. Male identity explained either no or only very small proportions of the variation in offspring or brood size. Larger males were weakly associated with larger offspring, but this effect was minimal (partial $R^2 \sim 1.5\%$). However, these patterns did not hold consistently across all data exclusion criteria and analytical variants, underscoring their tentative nature and highlighting the need for further investigation. Our study offers one of the first empirical tests of male effects in a gynogenetic vertebrate, providing valuable quantitative benchmarks for the magnitude of such effects in gynogenetic systems.

**Data availability statement:** The data underlying this study are available on Figshare (https://doi.org/10.6084/m9.figshare.31038220).

**Funding:** We received funding from the Deutsche Forschungsgemeinschaft (DFG) under Germany's Excellence Strategy EXC 2002/1, 'Science of Intelligence' (project number #390523135) and DFG 'Eigene Stelle' grant to SME (#536703956).

**Competing interests:** The authors have declared that no competing interests exist.

## Introduction

Parental effects – defined as any influence of parents on offspring phenotype that is not mediated by the transmission of alleles [1–3] – play an important role in shaping phenotypic variation across generations. While research on maternal effects has long dominated this field, receiving substantial empirical and theoretical attention [4–6], paternal effects have historically been overlooked [3,7]. For decades, they were largely dismissed as relevant only in species where males provide direct care [8,9]. However, a growing body of research now challenges this view, revealing that fathers can influence their offspring's phenotype even in the complete absence of paternal care. Thus, paternal effects are potentially more widespread and more influential than previously assumed, with recent evidence spanning taxa from insects to mammals [1,8,10] – even in species where males never interact with their young.

Aside from paternal care, males can shape offspring phenotypes through two other major non-genetic pathways. First, males may shape offspring phenotype via ejaculate-mediated effects. The ejaculate is comprised of both sperm and non-sperm seminal fluid [11]. Sperm can convey epigenetic information through mechanisms such as DNA methylation, histone modifications, or small non-coding RNAs, all of which can shape gene expression in the offspring without changing the underlying genetic code [11,12]. Non-sperm components of the seminal fluid – such as proteins, hormones, and lipids – may further shape fertilisation success and offspring phenotypes either by directly affecting embryonic development (e.g., growth, mortality) or indirectly, by influencing the maternal environment (e.g., stimulating egg maturation, providing nutrition, controlling pH, regulating female immune response) or through effects on the sperm (e.g., facilitating mobility, nourishing and protecting sperm) [11,13–20]. Importantly, effects of the seminal fluid on offspring can operate even when fertilization does not occur. This means even ejaculate from non-siring males can modify offspring phenotypes through the above-mentioned mechanisms [11,12]. For example, [21] mated female flies (*Telostylinus angusticollis*) with two males of differing condition and found that, although offspring were mainly sired by the second male, it was the condition of the first male that affected their body size. And in the cricket, *Teleogryllus oceanicus*, females were mated with both a sperm-donating male and a sterilized male. The authors of this study found that variability in offspring survival was linked to the sterilized male, which only contributed seminal fluid to reproduction [8]. Such findings highlight that ejaculate-mediated paternal effects are not restricted to genetic sires and that they can contribute to offspring variation in species with multiple matings as well as in gynogenetic species, where male genetic material is not incorporated into the offspring's genome. Second, an indirect link between males and offspring phenotypes can be created by females through differential resource allocation in response to male attractiveness [1,22–25]. For example, in a study on guppies, *Poecilia reticulata*, females produced more male-biased broods when mated with males who were experimentally manipulated to appear less attractive by presenting them alongside a more attractive stimulus male [26]. In zebra finches, *Taeniopygia guttata*, females mated to males that wore a red instead of a green leg ring (and hence, seemed more attractive) produced larger eggs [27].

In the present study, we use a powerful system – a gynogenetic unisexual vertebrate – to experimentally isolate non-genetic paternal effects from effects that are caused by the integration of male genetic material: the Amazon molly (*Poecilia formosa*). This gynogenetic freshwater fish emerged from a single hybridization event between the Atlantic molly (*Poecilia mexicana*) and the sailfin molly (*Poecilia latipinna*) estimated between 120,000–280,000 years ago [28–30]. As a gynogenetic species, Amazon mollies require sperm from one of the two above-named parental species or other closely related poeciliid species to trigger embryonic development [31]. Although oocyte and sperm interact and fuse, as demonstrated by rare instances of paternal introgression, the male's genetic material is generally not incorporated into the offspring's genome [32–34]. Because males do not contribute genetically to the next generation, paternity in the strict sense does not occur. Paradoxically, this very absence of genetic paternity makes the Amazon molly an ideal model to test for non-genetic paternal effects, as any male influence on offspring phenotype must arise independently of inherited alleles. Moreover, there is no parental care. As such, any male influence on offspring phenotype can only act through components of the ejaculate itself or through the effects of male-female interaction. However, up to now, the potential role of non-genetic paternal effects in Amazon molly reproduction has not been assessed (but see [31] for a recent study testing the effects of male geographic context (sympatric vs. allopatric) and genera (*Poecilia*, *Limia*, *Gambusia*, *Heterandria*, *Poeciliopsis*, *Xiphophorus*) on female reproductive success in the Amazon molly).

Here, we test for non-genetic paternal effects of Atlantic molly males on offspring phenotype (offspring size at birth) in Amazon mollies. Over 34 weeks (i.e., eight months), using tightly controlled laboratory breeding, we paired 60 Atlantic molly males with 54 Amazon molly females in a fully randomized, repeated measures design, resulting in 128 broods and 2,435 offspring. While females were standardized for age, size, descent, and developmental experience to minimize maternal variation, males were randomly drawn from a diverse stock population, thereby preserving natural phenotypic variation among males. We asked (i) whether male identity predicts offspring size at birth and (ii) whether male body size, often used as a proxy for male quality [35,36], predicts offspring size at birth. We expected male identity or body size might be associated with offspring size through potential direct or indirect effects of the ejaculate on embryonic development, or through differential female allocation in response to male phenotype (see above). Body size is one of the most fundamental phenotypic traits, with well-established links to individual fitness, e.g., through its effects on survival, development, competitive performance, and reproductive output [37–41]. Investigating paternal effects on offspring body size therefore provides critical insights into how male phenotypes may shape offspring quality and developmental trajectories. Beyond this, we also tested whether male identity or body size influenced brood size. While brood size itself is not an offspring trait, and thus not a paternal effect in the strict sense, it remains biologically relevant: brood size and offspring size typically trade off, as larger broods often come at the cost of smaller individual offspring [29–33]. Any influence of male identity on brood size could therefore represent an indirect pathway through which males affect offspring phenotypes [17]. Because Amazon mollies can store sperm [42] and were housed with multiple males in succession, we considered potential effects of both the male most likely to have triggered embryogenesis (hereafter 'primary male') and the male second most likely to have triggered embryogenesis (hereafter 'secondary male') on offspring size and brood size.

## Methods

### Reproductive cycle and sperm use in Amazon mollies

Amazon mollies are live-bearing and, like other poeciliid females, they follow a reproductive cycle characterized by a brief fertile period of about 2–3 days, occurring at the onset of sexual maturity and right after parturition, i.e., right after giving birth. This fertile period is followed by a roughly 30-day period during which females are not receptive to sperm. Fertilized eggs are carried internally until parturition, with gestation taking approx. 30 days [43,44]. Females can store sperm and spermatozoa are present in the ovarian epithelium during both the non-gestation and gestation phases [42]. We note that, in a related poeciliid fish [45], an extreme fertilization bias towards fresh sperm has been reported. In particular, guppy females (*P. reticulata*) were first artificially inseminated with sperm from one male and after parturition, were then

inseminated with sperm from a different male: in the subsequent brood, the fertilization bias towards the fresh sperm was 98.6 ± 5.2% (mean ± SD). However, it is currently unclear whether such fresh-sperm precedence operates to the same degree in the Amazon molly.

## Holding conditions

Atlantic molly males and Amazon molly females used in our experiment originate from stock populations kept at Humboldt-Universität zu Berlin (Berlin, Germany). All fish were housed according to the recommendations outlined in [46], i.e., stock populations were kept in 200L tanks with approx. 50 individuals per tank under the following holding conditions: 12:12h light:dark cycle, air temperature control at approx. 24 ± 1°C, weekly water changes. Fish were fed twice a day for 5 days a week with commercial powder food (Sera vipan baby).

## Experimental procedure

To launch our experiment, we placed $N = 66$ virgin, juvenile Amazon molly females (age = 42 days) – standardized for descent and experiential background (see 'Female standardisation' below) – into highly standardized conditions, that is, identical, individual 11L housing tanks equipped with Sera Biofibres for refuge and standardised feeding protocol, where they remained throughout the study. When females reached 70 days of age, we started the breeding period (reproductive onset was observed to occur around this time in our previous work [47]. Females were allowed to reproduce until they were 260 days old. Males ($N = 67$), randomly selected from a *P. mexicana* stock population as to reflect a natural range of phenotypic variation, were uniquely tagged with VIE elastomer markers for identification and then transferred to individual housing tanks (same as female tanks) at least 7 days before their first pairing. While females remained in their initially assigned tanks, males were rotated among them (see below), allowing us to experimentally control for female identity and potential tank effects on reproductive characteristics at the same time.

Breeding was conducted in two phases (Fig 1): a pre-reproductive phase (Phase 1) and a reproductive phase (Phase 2). The key difference between these phases lies in the timing and duration of male presence, as well as female reproductive status, i.e., Phase 1 covers the period up to a female's first parturition, while Phase 2 starts immediately after parturition of the first brood and covers all subsequent reproductive events. In Phase 1, each female was continuously housed with a male as to guarantee access to sperm during her fertile window, the timing of which was not yet known (see 'Study species and holding conditions' for details regarding female reproductive cycle). Thus, every two weeks an unfamiliar male (i.e., a male not previously paired with the female) was introduced into a female's tank and remained there until the next male was rotated into the tank. Once a female produced her first brood (age at first parturition for the $N = 54$ females

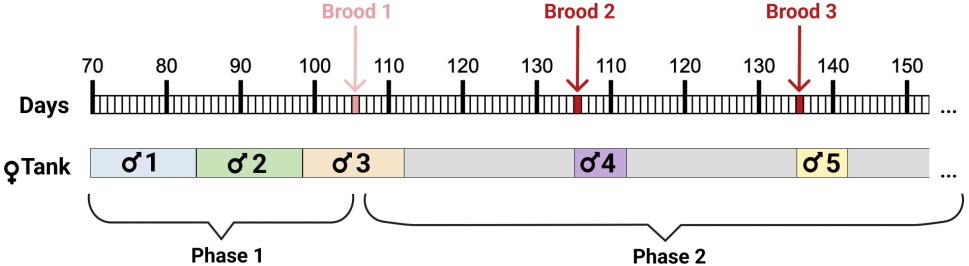

**Fig 1. Schematic illustration of our breeding protocol for one exemplary female.** Amazon molly females (*Poecilia formosa*) were sequentially paired with unfamiliar *Poecilia mexicana* males in two breeding phases. In Phase 1 (pre-reproductive), males were rotated every two weeks until females produced their first brood (light red arrow). In Phase 2 (reproductive phase), a new unfamiliar male was introduced at the birth of each brood (indicated by dark red arrows) and remained with the female for seven days to ensure insemination; for first broods, this role was fulfilled by the male already present in the tank. Brood 1 is shown for completeness only and is not included in the main analyses.

in our final dataset is 181.1±39.8 days, mean±SD), we initiated Phase 2. In Phase 2, we could accurately identify when a female was receptive, namely, directly after she gave birth. At these specific times, a new unfamiliar male was introduced for a shorter but sufficient period of seven days to trigger the development of the next brood. At the transition from Phase 1 to Phase 2, females already had a male in their tank (as males were continuously present during Phase 1), accordingly, this male remained for an additional seven days after the first brood to ensure insemination. While Phase 1 was necessary to effectively initiate reproduction, the transition to Phase 2 allowed us to minimize confounding effects – both from overlapping male exposures during a female's fertile window and from potential behavioral interactions between males and females during gestation.

We checked all tanks for broods twice a day for 7 days a week. Once we found a brood, all offspring were taken out, counted and measured for total length (length from the tip of the snout to the end of the tail fin). Female total length was also measured on the day of parturition. Additionally, we measured females for total length every time a male was introduced into their tank during Phase 1 (16.1±0.2 size measurements per female, mean±SE=3.067±1.601 cm, $N$=853 size measurements from 54 females included in the final dataset). Males were measured for total length every time they were transferred to or from a female's tank (10.3±0.6 size measurement per male, mean±SE=3.497±0.403 cm, $N$=612 size measurements from 60 males included in the final dataset). Body sizes were measured using our custom-developed software [48]. For $N$=1 brood, offspring sizes are missing and for $N$=2 broods, brood sizes are missing. $N$=4 females did not reproduce.

Each fish received 1/64 tsp of Sera vipan baby power food twice a day for 7 days a week, i.e., tanks with both a male and a female fish received 1/32 tsp and individual fish received 1/64 tsp food per feeding. Breeding tanks were visually separated from each other and spread over two recirculating tank systems ($N$=51 and $N$=56 tanks per system, respectively). After the experiment, all fish returned to the fish husbandry, i.e., we did not sacrifice any fish.

## Male assignment to broods and exclusion of Phase 1

For each brood, the male who was in the female's tank 30 days before parturition was identified as the 'primary male'. Given the strong fertilization bias towards fresh sperm observed in a closely related poeciliid (see 'Study species and holding conditions'), this male most likely triggered embryonic development of the brood in question. For example, in Fig 1, male number 4 would be identified as the primary male triggering the development of Brood 3. As females can store sperm [42], we additionally accounted for potential effects arising from multiple males' ejaculates being present in the female's reproductive tract by identifying the male housed with the female directly before the primary male as the 'secondary male'. For example, in Fig 1, male number 3 would be identified as the secondary male triggering the development of Brood 3. We did not identify third and fourth males because our sample size was limited and because we expected more distant encounters with males to be of less importance. For $N$=6 broods, the primary male could not be assigned with certainty because no brood was recorded 30 days prior to parturition and, consequently, there was no male was present in the female's tank at that time. These broods were excluded from our final dataset.

We excluded Phase 1 of reproduction (i.e., each female's first brood) from our final dataset due to higher uncertainty in reliably assigning broods to individual males. Frequent male swappings and the continuous presence of males in female tanks often resulted in overlapping exposure during the females' receptive periods. For example, for $N$=20 broods in Phase 1, two males were present in the female's tank 30±2 days prior to parturition. Furthermore, after being housed with the primary male, females were housed with two additional males during gestation (Fig 1), rendering this phase less standardized with respect to potential behavioral effects and complicating any inference regarding male identity.

The full dataset (including Phase 1 data) was comprised of $N$=3,193 offspring from $N$=192 broods. Our final dataset (excluding Phase 1) included $N$=2,435 offspring and $N$=127 broods. On average, females contributed 2.4±1.1 broods to our final dataset, and primary males were assigned to 2.5±1.5 broods, secondary males were assigned to 2.3±1.4 broods.

## Female standardization and prior experimental experience

Following 3R guidelines that maximize data collected from individual animals, females used in this experiment had previously (i.e., during the first 42 days of their life) participated in a different experiment (manuscript in preparation). We here briefly describe this experiment to clarify the origin and standardization of females. In this prior experiment, we tracked female behavior over the first six weeks of their life under highly standardised conditions (i.e., identical and individual tanks from day 1 of their life, standardised feeding protocol). For a duration of 2 weeks within this six-week window, half of the females were subjected to a predator treatment, i.e., they were presented with olfactory predator cues (*Crenicichla acutirostris*) and were chased for 2 min per day with a pike cichlid model. For the other half of the females, experimental conditions remained constant (control treatment). The prior experiment was run in $N = 3$ experimental blocks with approx. six weeks in between blocks. Females were standardized for their descent, i.e., they originated from $N = 6$ mothers, all of which were full-siblings (i.e., derived from a single brood), in the following referred to as female origin. That means, all females originate from a single lineage. All females were transferred to individual breeding tanks immediately after the prior experiment, i.e., when they were 42 days old.

In the present study, to control for potential carry-over effects of prior female experimental usage, we controlled all statistical models for potential effects of treatment (predator vs. control), experimental block (1–3) and female origin (mother 1–6) on female reproduction (see 'Statistical analyses'); we note that none of these factors significantly affected offspring sizes or brood sizes (see S1 File).

## Statistical analyses

Statistical analyses were performed in R version 4.2.1 [49] using the following packages: *arm* [50], *dplyr* [51], *ggplot2* [52], *ggpubr* [53], *lme4* [54], *partR2* [46] and *sjPlot* [55]. For all models, model assumptions were verified using residual and q-q plots. Following [56,57], we did not employ a model selection procedure; instead, our results were obtained from full models.

To test for an effect of male identity on the size of offspring and broods produced, we built two linear mixed-effects models (LMM) with either offspring size ($N = 2,435$ offspring from $N = 127$ broods) or brood size ($N = 126$ broods) as the response. Female size at parturition (predicted from individual von Bertalanffy growth curves, [58]), tank system, tank level, and tank centrality were included as covariates. Females were housed in two different tank systems, with individual tanks being distributed over four levels. Within these systems, tank centrality was classified as either central (with neighboring tanks on both sides) or peripheral (edge-tank with only one neighboring tank). We controlled for potential carry-over effects from female prior experimental usage by further including female treatment (predator vs. control) and experimental block (1–3) as covariates (see 'Female standardization and prior experimental experience' above). As random terms, we included primary and secondary male identities, female/tank identity (each female was housed in a single tank, making tank identity and female identity synonymous, see 'Experimental procedure'), female origin, and brood identity (offspring model only). We did not control for female age but female size at parturition; as age and size were tightly correlated (linear mixed-effects model with predicted female size at parturition as response, female age as predictor, and female/tank identity as random term, $N = 128$ broods from $N = 54$ females: $\chi^2 = 230.16$, $p < 0.001$, marginal $R^2 = 0.703$, conditional $R^2 = 0.946$). Our models include male and female identities as random effects, but not their interactions, since in our experimental design each male was paired with a female only once, resulting in a single brood per unique male–female combination. Without repeated measurements for these combinations, male-female interactions cannot be meaningfully estimated. Following [59], we calculated repeatabilities (R) with 95% confidence intervals (CI) for both primary and secondary males as the variation explained by among-male differences divided by the total variation in the population. Therefore, repeatability refers to the proportion of the variation in offspring or brood size, respectively, that can be attributed to differences among males. Significance is derived from 95% CIs not overlapping with zero.

To test for an effect of male size on reproductive characteristics, we added male total length (averaged over all individual size measurements) as a predictor to both above-described models. We used average male length rather than brood-specific male length, as poeciliid males cease, or at least markedly reduce, somatic growth upon reaching sexual maturity [60–62]. Variation in repeated size measurements of the same individual is therefore more likely to reflect measurement error rather than true growth. Consequently, we considered average length the more robust measure (see S1 Fig in S1 File for an overview of male size measurements).

## Use of AI-assisted tools

The AI-based language tool ChatGPT 5.2 was used to assist with language editing and stylistic refinement. All scientific content, including data analysis, and conclusions were developed by the authors.

## Results

Primary and secondary male identity explained either no or only very small proportions of the variation in the size of offspring or broods produced, i.e., repeatability estimates were 2% at best (Table 1; Fig 2). Furthermore, we found a statistically significant, yet weak, correlation between male body size and offspring size for both primary and secondary males: larger males were associated with slightly larger offspring (primary male: p-value=0.011, secondary male: p-value=0.034; S1 Table in S1 File, Fig 3a-b). However, the effect was vanishingly small, with males explaining only 1.5% (partial $R^2$ primary male) and 1.1% (partial $R^2$ secondary male) of the variance. In contrast, male size (primary or secondary) did not predict brood size (primary male: p-value=0.475, secondary male: p-value 0.124; S2 Table in S1 File, Fig 3c-d).

Phase 1 data were excluded from our main analysis above, as frequent male rotations caused overlapping exposures during females' receptive periods (see 'Male assignment to broods and exclusion of Phase 1' for more details) and decreased environmental standardisation during gestation. Nevertheless, our results above were largely robust towards the inclusion of non-overlapping Phase 1 broods, i.e., broods where females were exposed to a single male during their fertile window (30±2 days prior to parturition; $N=42$ broods). When including these broods, we continued to find no or only very small effects of male identity on either offspring or brood size (Table 1) and no effect of male size on brood size

Table 1. Repeatabilities (with 95% confidence intervals) for offspring size and brood size, attributed to primary and secondary males across different analytical scenarios. The main analysis presents repeatabilities estimated from the final dataset, while subsequent columns present the robustness of these estimates when (i) including Phase 1 reproduction, (ii) excluding males who contributed only a single brood to the final dataset, and (iii) excluding old or poor-condition males. Sample sizes (number of offspring, broods, and males) are shown for each analysis.

| | | | Main analysis | Robustness of the main analysis with respect to: | | |
| --- | --- | --- | --- | --- | --- | --- |
| | | | | Inclusion of Phase 1 reproduction | Exclusion of single-brood males | Exclusion of old/ poor condition males |
| Offspring size | R with 95% CI | Primary male | 0.000 [0.000, 0.000] | 0.000 [0.000, 0.000] | 0.000 [0.000, 0.000] | 0.069 [0.044, 0.105] |
| | | Secondary male | 0.002 [0.001, 0.003] | 0.022 [0.015, 0.033] | 0.000 [0.000, 0.000] | 0.035 [0.022, 0.053] |
| | N | Offspring | 2,435 | 2,966 | 1,733 | 1,596 |
| | | Broods | 127 | 169 | 90 | 82 |
| | | Primary males | 50 | 59 | 33 | 39 |
| | | Secondary males | 56 | 58 | 33 | 41 |
| Brood size | R with 95% CI | Primary male | 0.022 [0.015, 0.032] | 0.000 [0.000, 0.000] | 0.118 [0.070, 0.172] | 0.057 [0.036, 0.083] |
| | | Secondary male | 0.000 [0.000, 0.000] | 0.000 [0.000, 0.000] | 0.000 [0.000, 0.000] | 0.000 [0.000, 0.000] |
| | N | Broods | 126 | 168 | 90 | 81 |
| | | Primary males | 50 | 59 | 33 | 38 |
| | | Secondary males | 56 | 58 | 33 | 41 |

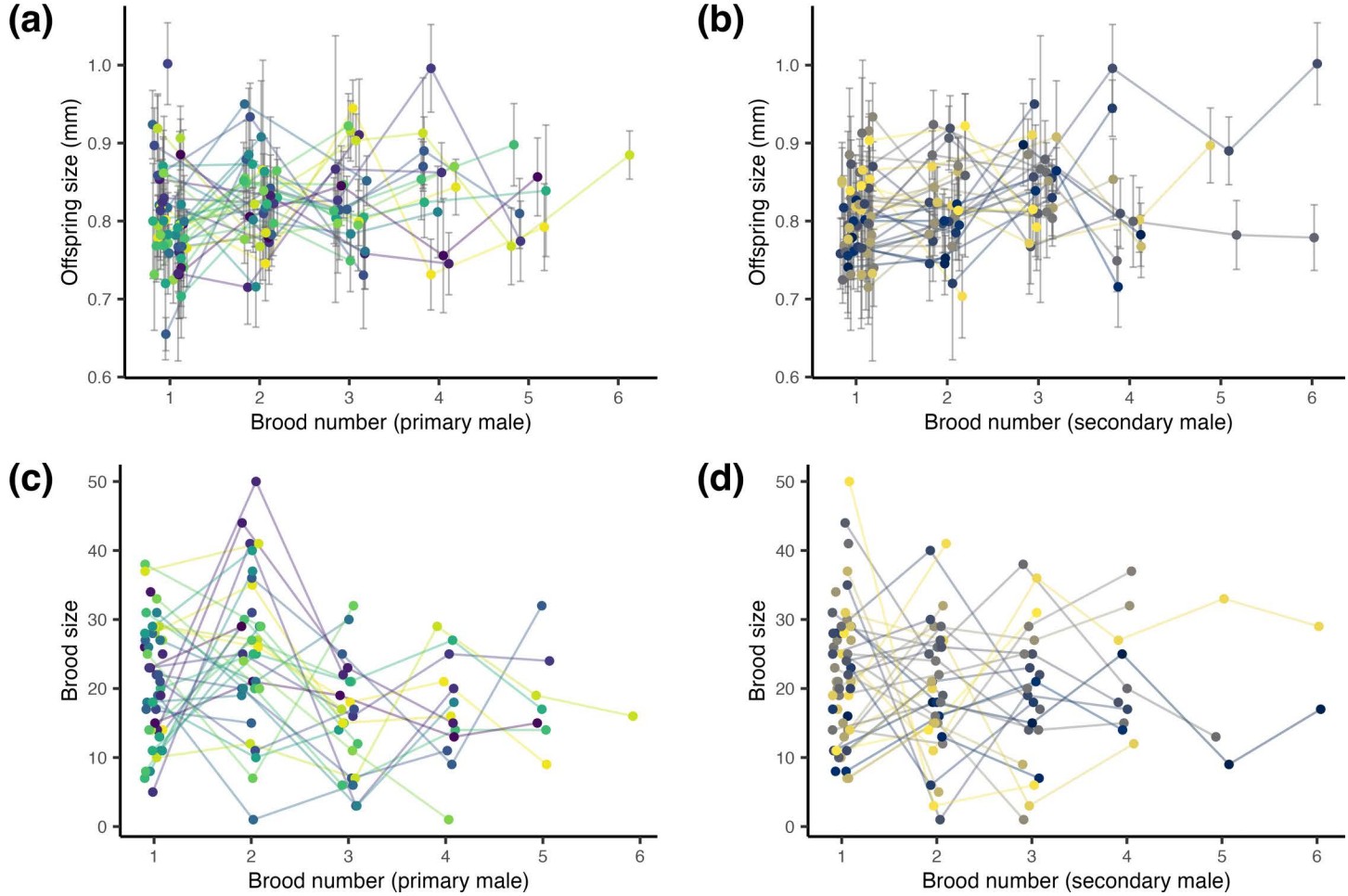

**Fig 2. Relationships between male identity and reproductive characteristics in Amazon molly broods.** Panels **(a)** and **(b)** show mean offspring size per brood (*N* = 2,435 offspring from *N* = 127 broods) for primary and secondary males, respectively. Error bars represent within-brood standard deviations. Panels **(c)** and **(d)** show brood sizes (*N* = 126 broods) for primary and secondary males, respectively. **(a-d)** Primary males refer to males housed with females immediately prior to brood production and are therefore most likely to have triggered embryogenesis, whereas secondary males refer to males housed with females before primary males and are therefore second most likely to have triggered embryogenesis. Each point represents one brood, and each line represents one male (both colored by male identity). Males were assigned to at least one, but up to six broods, broods are numbered chronologically per male.

(primary male: p-value = 0.672, secondary male: p-value = 0.392; S4 Table in S2 File). Except, the previously reported weak link between male size and offspring size became non-significant (primary male: p-value = 0.178, secondary male: p-value = 0.191; S3 Table in S2 File).

All males were included in our main analysis above, including those who contributed only a single brood (single-brood males: *N* (primary males) = 17, *N* (secondary males) = 23). Although males with repeated reproductive events contribute most meaningfully to repeatability estimates – by allowing reliable partitioning of within- and among-individual variance – single-brood males still add valuable information on population-level means and variances, ensuring that the full spectrum of reproductive outcomes is represented. However, estimates based on males with only a single recorded brood may be less reliable, as their average trait value cannot be robustly determined. To account for this potential bias, we conducted a robustness analysis excluding single-brood males from our final dataset. When doing so, male repeatability estimates for

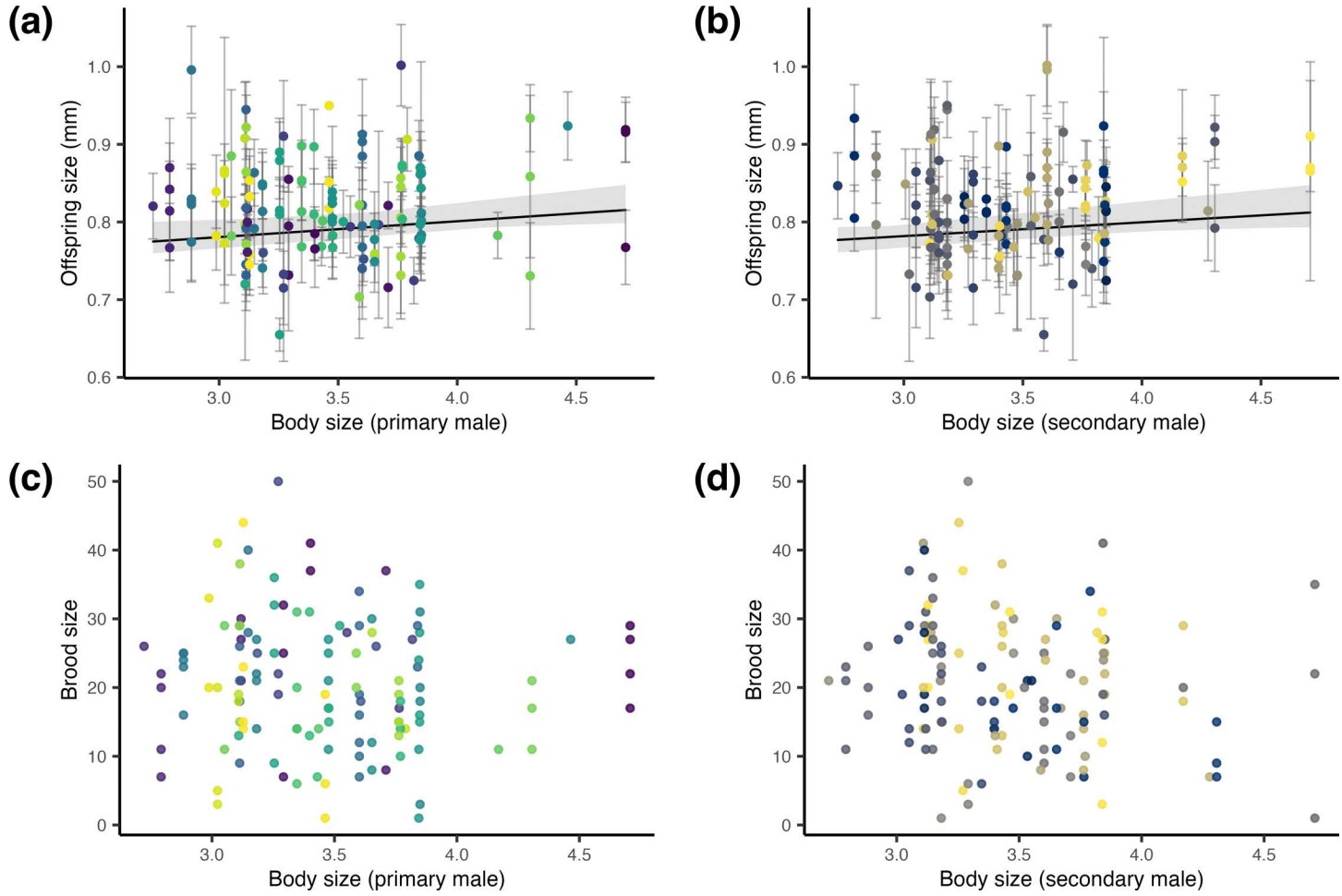

**Fig 3. Relationships between male size and reproductive characteristics in Amazon molly broods.** Panels (a) and (b) show the relationship between male size and mean offspring size (*N*=2,435 offspring from *N*=127 broods) for primary and secondary males, respectively. Solid lines and shaded areas indicate the fitted regression lines with 95% confidence intervals. Error bars represent within-brood standard deviations. Panels (c) and (d) show the relationship between male size and brood size (*N*=126 broods) for primary and secondary males, respectively. **(a-d)** Primary males refer to males housed with females immediately prior to brood production and are therefore most likely to have triggered embryogenesis, whereas secondary males refer to males housed with females before primary males and are therefore second most likely to have triggered embryogenesis. Each point represents one brood, colored by male identity.

offspring size remained qualitatively consistent, while repeatability for brood size increased notably, reaching roughly 12% (Table 1). The results of our models testing for effects of male size on offspring or brood size were largely unchanged: the previously observed weak association between male size and offspring size persisted, although now driven only by the primary male (primary male: p-value=0.023, partial $R^2$=0.017; secondary male: p-value=0.251; S5 Table in S3 File). There was no link between male size (primary or secondary) and brood size (primary male: p-value=0.344, secondary male: p-value=0.211; S6 Table in S3 File).

As our study was conducted over a period of ten months (with the typical lifespan of poeciliid fish in the lab being ~2 years, *personal observation*), and as males of unknown ages were sampled from the stock population, some males died during the experiment (*N*=23 males). To test for potential effects of age or condition, we repeated our main analyses excluding all broods and offspring assigned to males (primary and secondary) who died within two months after mating. In

this analysis, primary male identity explained 5.3% of the variation in brood size whereas secondary males continued to show no effect (Table 1). For offspring size, the proportion of explained variation increased to 6.9% (primary males) and 3.5% (secondary males) (Table 1). The weak association between male size and offspring size persisted, although it was attributable only to primary males (primary male: p-value = 0.006, partial $R^2$ = 0.020; secondary male: p-value = 0.451; S7 Table in S4 File) and no link between male size and brood size was observed (primary male: p-value = 0.648, secondary male: p-value = 0.286; S8 Table in S4 File).

Following [56,57], in our main analysis above, we did not employ a model selection procedure when testing for links between male body size and offspring or brood size. However, when selecting most parsimonious models via backward elimination of non-significant terms, the weak association between male size and offspring size was no longer significant (primary male: p-value = 0.064, secondary male: p-value = 0.069; S9 Table in S5 File). The absence of a link between male size and brood size remained, although we detected a strong trend for the secondary male during the model selection process (p-value = 0.050; S10 Table in S5 File).

We note that, in line with previous results [47], female identity explained a significant proportion of the variation in the size of offspring (repeatability with 95% confidence interval = 0.178 [0.129, 0.240]) and broods (repeatability with 95% confidence interval = 0.164 [0.115, 0.216]) produced. However, because each female was housed in a single tank, we cannot conclusively isolate effects of female identity from potential tank effects.

While we report only p-values here, complete model summaries for all main and supporting analyses (including estimates ± SE and test statistics for all fixed effects and covariates) are provided in S1–S5 File.

## Discussion

In this study, we used the gynogenetic Amazon molly – a naturally clonal vertebrate that requires sperm to trigger embryogenesis but excludes male DNA – to test whether males can influence offspring phenotypes in the absence of genetic inheritance and parental care. Leveraging a highly controlled experimental setup and a robust repeated-measures design, we asked whether (i) male identity or (ii) male size predicts offspring size at birth. We further tested for effects of male identity and/or size on brood size. Overall, we found weak evidence for non-genetic paternal effects: male identity explained no or only very little of the variation in offspring or brood size; and while larger males were statistically associated with slightly larger offspring, the effect size was minimal, explaining only up to ~1.5% of the variance. However, these results were not robust towards all data exclusion criteria and modelling variations, emphasising the tentative nature of these patterns and the need for continued investigation.

Although we found an association between male body size and offspring size, it was minimal and vanished when applying backwards model selection or when analyzing the full dataset including Phase 1 data. Similarly, consistent among-male differences in offspring and brood size emerged only after excluding N = 45 broods from males who died within two months after the mating from our final data set; or after excluding single-brood males (i.e., after excluding N = 37 broods from our final dataset; brood size only). The instability of our findings across different data exclusion criteria and analytic variants suggests that male influences on the size or number of offspring produced in Amazon mollies cannot be entirely dismissed, but if present, they are likely weak. In other words, while our data provide no compelling evidence for paternal effects in our system, they also do not conclusively demonstrate their absence.

Several non-exclusive explanations may account for the limited evidence observed. First, males were drawn from a single stock population, with little environmental variation among them. If paternal effects arise primarily through environmentally mediated mechanisms, such as stress- or food-induced changes in ejaculate composition [18,63–65], then using males reared and maintained under standardised conditions may have minimized variation in the very traits that could influence offspring phenotype or fertilisation success. Future studies should thus experimentally manipulate male condition, diet, or stress exposure to test whether environmental heterogeneity amplifies paternal effects in this system. Such work would provide valuable insight into the ecological relevance of paternal effects in Amazon mollies under more

natural conditions and within wild populations. Second, even though we analysed nearly 2,500 offspring and 128 broods, the effective sample size for estimating among-male variation may have been modest – particularly when the expected magnitude of effect is small – because statistical power in this context depends primarily on the number of unique males and their number of repeated measures rather than on the total number of offspring. In our study, 60 males contributed on average roughly 2.5 broods each, with several males represented by only a single brood. This limited replication per male likely constrained our ability to detect subtle paternal effects. Considerably larger datasets with more repeated reproductive events per male may be required to reliably detect weak male influences in this system. Third, it remains possible that paternal effects manifest in traits other than offspring size, e.g., offspring metabolism, stress reactivity or behavior [19,65–67]. For example, [65] showed in a study on mice how differences in male cortisone signalling via seminal fluid affects metabolic health and feeding behavior in male offspring. Also, a study on European whitefish, *Coregonus lavaretus*, demonstrated the effects of seminal fluid on embryo survival, hatching time, and offspring swimming performance [68]. Exploring paternal effects across a broader range of offspring traits could reveal more nuanced ways in which males affect offspring phenotypes in the Amazon molly.

A further limitation of our study is that all females originated from a single clonal lineage. While this design ensured complete genetic uniformity among females, allowing us to standardise female variation – it also constrains the generality of our results. Our findings therefore apply specifically to this lineage and should be interpreted within that genetic context. It remains possible that other Amazon molly clonal lines, differing in their genetic background or evolutionary history, may differ in their sensitivity to paternal influences. Moreover, the patterns observed here may not necessarily extend to other gynogenetic or sexually reproducing species, where reproductive physiology, sperm-egg interactions, or ejaculate composition may differ substantially. Future work incorporating multiple clonal lineages and closely related species will thus be essential to determine whether the magnitude or presence of paternal effects depends on genotype, population history, or reproductive mode.

In conclusion, our study provides one of the first empirical assessments of non-genetic paternal effects in a gynogenetic vertebrate, and our results therefore contribute valuable quantitative benchmarks for the scale of such effects in gynogenetic systems. Our findings suggest that any male influences on offspring and brood size, if present, are likely subtle and context-dependent. We therefore encourage future research to build on this foundation by employing larger sample sizes with more repeated reproductive measures per male, experimentally manipulating male condition, and exploring additional offspring traits that may be more sensitive to paternal contributions. In doing so, we may gain a better understanding of the conditions under which paternal phenotypes can shape offspring outcomes in the absence of genetic input.

## Supporting information

**S1 File. Model summaries & supporting figures.**
(PDF)

**S2 File. Robustness analysis with respect to the inclusion of Phase 1 reproduction.**
(PDF)

**S3 File. Robustness analysis with respect to the exclusion of single-brood males.**
(PDF)

**S4 File. Robustness analysis with respect to the exclusion of presumably old or poor-condition males.**
(PDF)

**S5 File. Robustness analysis with respect to selecting most parsimonious models.**
(PDF)

  

## Acknowledgments

Members of the lab at the Humboldt University of Berlin and the Leibniz Institute of Freshwater Ecology and Inland Fisheries provided invaluable help with animal husbandry, experimental routine, data acquisition and processing; particular thanks are extended to Christopher Schutz and Ronja Leipoldt.

## Author contributions

**Conceptualization:** Ulrike Scherer, Sean M. Ehlman, David Bierbach, Jens Krause, Max Wolf.

**Data curation:** Ulrike Scherer.

**Formal analysis:** Ulrike Scherer.

**Funding acquisition:** Sean M. Ehlman, Jens Krause, Max Wolf.

**Investigation:** Ulrike Scherer.

**Methodology:** Ulrike Scherer, Sean M. Ehlman, David Bierbach, Jens Krause, Max Wolf.

**Writing – original draft:** Ulrike Scherer, Max Wolf.

**Writing – review & editing:** Ulrike Scherer, Sean M. Ehlman, David Bierbach, Jens Krause, Max Wolf.

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
