## [Decision Letter · Decision Letter 0]

22 Aug 2025

Dear Dr. Scherer,

Thank you for submitting your manuscript to PLOS ONE. After careful consideration, we feel that it has merit but does not fully meet PLOS ONE’s publication criteria as it currently stands. Therefore, we invite you to submit a revised version of the manuscript that comprehensively addresses the points raised during the review process. Note that both reviewers have made a number of highly pertinent comments that require particular attention during revision of the manuscript.

We look forward to receiving your revised manuscript.

Kind regards,

Michael Schubert

Academic Editor

PLOS ONE

2. To comply with PLOS ONE submissions requirements, in your Methods section, please provide additional information regarding the experiments involving animals and ensure you have included details on (1) methods of sacrifice, and (2) efforts to alleviate suffering.

[We received funding from the Deutsche Forschungsgemeinschaft (DFG) under Germany's Excellence Strategy EXC 2002/1, ‘Science of Intelligence’ (project number #390523135) and DFG ‘Eigene Stelle’ grant to SME (#536703956).].

Reviewers' comments:

Reviewer's Responses to Questions

**Comments to the Author**

1. Is the manuscript technically sound, and do the data support the conclusions?

Reviewer #1: Yes

Reviewer #2: Partly

2. Has the statistical analysis been performed appropriately and rigorously?

Reviewer #1: Yes

Reviewer #2: Yes

3. Have the authors made all data underlying the findings in their manuscript fully available?

Reviewer #1: No

Reviewer #2: No

4. Is the manuscript presented in an intelligible fashion and written in standard English?

Reviewer #1: Yes

Reviewer #2: Yes

Reviewer #1: This manuscript presents a study examining paternal effects in the gynogenetic Amazon molly. The research question is intriguing and relevant, and the manuscript is generally well written. However, I have several concerns regarding both methodology and analysis that need to be addressed to strengthen the validity of the findings. The core issue lies in the interpretation of paternity in a system where males do not contribute genetically, raising questions about the assignment of paternal identity. Additionally, there are analytical and reporting elements that require clarification or revision.

Major Comments:

1. Interpretation of “Paternity” in Gynogenetic Species:

It is problematic to frame the analysis around the assignment (or lack thereof) of paternity to specific males when these males do not contribute genetic material. I recognise that this is probably a simpler way to frame it, to call it sired, fathered, etc. rather than calling it the identify of males that were housed with females, but it’s not technically accurate? Also, as females were exposed to multiple males across time, stored sperm may influence brood outcomes, meaning the most recent male may not be the one affecting brood traits. While the manuscript cites evidence for strong fertilization bias, this comes from a different species (P. reticulata), which is not gynogenetic. It is unclear whether sperm precedence operates similarly in the Amazon molly. The authors should clarify this assumption or provide more robust justification for its applicability here, and address this potential limitation in the discussion.

2. Tank Effects:

The manuscript mentions a test for “tank system” but does not appear to include individual tank identity (tank ID) as a factor. This is potentially problematic, as tank-level variation could confound the results. A random effect for tank ID should be included, or at minimum, tank effects should be explicitly tested and reported.

3. Inclusion of Males with Only One Brood:

Figure 1 indicates that some males sired only a single brood. What is the justification for including these males in the repeatability analysis? These single observations may not contribute meaningfully to the estimation of variance components and could dilute the signal. The authors should test whether excluding these cases changes the results and report accordingly. Likewise, there are a lot of factors in the models, many of which are not significant. Have you tried running he models without non-significant fixed effect/covariates? Though I note that it doesn’t seem like it will change the results based on the data plotted in the figures.

4. Random Effects Structure:

It is unclear why the interaction between male and female identity was not included as a random effect. Given the experimental design, such an interaction could capture potential dyad-specific variation in brood traits?

5. Experimental Design: It would help if you could state how often females received a new unfamiliar male - it was every two weeks, but for how long, continuously from 70 to 260 days old for some females? And this was just until they produced the 1st brood, yes, and then it was a new male after every new brood? Or did the 2-week cycle start again for this next brood? On line 138 it says that in phase 2, an unfamiliar male was added whenever she gave birth to a new brood, but this was 7 days after giving birth to the new brood, yes?

Minor Comments / Line-by-Line Suggestions:

• Line 54: Add references to support the statement: “…from insects to mammals.”

• Line 70: Clarify wording. The cited study did not find significant effects of male species; it reported a non-significant trend in the allopatric vs. sympatric comparison.

• Line 93: Replace “mollies” with “molly.”

• Line 118: Replace “juveniles” with “juvenile.”

• Line 133-138: Clarify the male exposure schedule. Was an unfamiliar male introduced every two weeks from 70 to 260 days of age, or just until the first brood? After the first brood, were females exposed to a new male each time they reproduced? Line 138 states that the new male was introduced seven days after parturition—this needs to be clearer and more consistent with earlier descriptions.

• Line 148: Remove “a” from “the day they gave a birth.”

• Line 178: The sentence “no effect of either factor” is followed by three factors. Please revise for consistency.

• Line 191: Remove the comma: change “(32), (33)” to “(32,33).”

• Line 197: Add “the” before “response.”

• Line 211-212: Why is average male length used? If brood-specific male length was measured, it would seem more appropriate and accurate to use that value.

• Line 217: Clarify that repeatability estimates of zero reflect zero variance explained by male identity in the models.

• Line 225: Explain the rationale for excluding males that died within two months. Was this to account for potential illness affecting reproductive interactions?

Figures and Supplementary Material:

• Figure 1: Please clarify why the figure shows a single data point per brood. Would it not be more informative to show average brood size or trait value, especially if multiple offspring were measured?

• Supplementary Material: Consider revising the layout of the random effects and associated sample sizes. Instead of using rows to indicate sample sizes, include the sample size alongside each random effect name for clarity.

Reviewer #2: This paper looks at the effects of paternal identity on offspring traits in a gynogenetic system, where males do not contribute genetically to the production of offspring. The authors found that male identity did not significantly predict offspring size or brood size. While the writing was clear and concise, I have a number of significant issues with the experimental design and manuscript.

1. We generally need more background about the mechanisms by which sperm can induce paternal effects, both when the egg is fertilized and when it is not. For example, what are typical mechanisms that fathers use to influence offspring and which ones would be potential mechanisms that could still occur even if the father did not contribute genetic material? Some of this background needs to be species specific. What happens to the sperm in this system- does it actually fertilize the egg (such that non-coding RNAs could influence gene expression in the developing embryo) or is the sperm just in the vicinity (such that proteins in seminal fluid might be important)?

2. I found the literature review and introduction to generally be a bit lacking in detail. There are a number of studies in the sperm competition literature about the role of a secondary male’s sperm in the vaginal tract and how it affects their non-biological offspring. This would be important to draw on. Including some of this information would also give a better background about why this is important. On one hand, this is a unique system for looking at these effects, but on the other hand, it is possible that these effects are not generalizable beyond this species.

3. Why those traits? Why would male identity affect offspring size or brood size? You partially address this, but those seem to be traits that would be much more likely to be linked to female body condition or size. Brood size is largely determined by female size, and then individual offspring size is a reflection of brood size. Indeed, if we looked at paternal effects in many species, we probably would see no effect of paternal identity on these traits (I don’t, in my system), yet we still see very strong effects of paternal experiences.

4. The methods are very unclear about the relationship of the females to each other. It seems like the females are all genetically identical to each other, rather than coming from separate lineages. Therefore, there is an issue with pseudoreplication- you can conclude that paternal effects do not matter (for the traits you measured) only in this one particular lineage, but it could be that this one particular lineage or population does not respond to males, but others do.

5. Generally, the result feel overstated on Line 279-291. While in many ways, the experimental design is neat, the authors have only ruled out the effects of male ID on two traits at birth, but not on traits later in life (which you state). Importantly, the authors didn’t look for the effects of paternal experiences by manipulating a stressor -just looked for the effects of male ID (which are different types of paternal effects). Given that this was tested in only one lineage, the results need to be much more cautious and the title of the paper should reflect that.

Minor comments

Line 20- I would consider this definition to be pretty far removed from the papers that are cited. It is unclear what is meant by the integration of genetic material- I would just say that that they are not related to the parental genotype.

Lines 133-143; it is unclear why there are phases to reproduction, this should be justified. In phase 1, did you get rid of the babies? Also, if part of the paternal effects could be triggered by behavioral interactions with the male, your experimental design, with each female interacting with multiple males, probably couldn’t fully detect this because investment in offspring could be the result of her interaction with the previous male, not the current male.

Line 137- what is pseudo-fertilization?

Line 153- need a citation or details about the custom software.

Lines 217-219- need statistical test for this repeatability and more details about how you set this up

**Do you want your identity to be public for this peer review?** For information about this choice, including consent withdrawal, please see our Privacy Policy

Reviewer #1: No

Reviewer #2: No

---

## [Author Response · Author response to Decision Letter 1]

12 Nov 2025

Dear Dr. Schubert,

We would like to sincerely thank you and the reviewers for the careful and constructive evaluation of our manuscript. The detailed feedback has been invaluable and has led to substantial improvements in the clarity, rigor, and scope of our work.

In our revision, we have comprehensively reworked and expanded the manuscript in response to all reviewer comments. Major changes include:

- A substantially deepened Introduction, now providing detailed background information on potential mechanisms and pathways through which paternal effects may arise in our system, including ejaculate-mediated effects and female differential allocation (lines 53-80).

- A thoroughly revised Methods section, providing clearer and more detailed descriptions of the experimental design and breeding procedures (lines 165-190 and lines 210-236).

- Analytical revisions, where we now account for the potential effects of multiple males (by distinguishing “primary” and “secondary” males), exclude Phase 1 reproduction from the main analyses due to reduced standardization and overlapping male exposures, and control for potential effects of tank position (level and centrality) (lines 269-282).

- Updated Results, reflecting these analytical refinements and presenting several robustness analyses. Notably, we now detect weak, non-genetic male effects, although these effects are small and not robust across all model variants (lines 305-402).

- A rewritten Discussion, where we substantially toned down our conclusions, emphasizing the tentative nature of our findings, and acknowledging key limitations like the use of a single clonal lineage, the limited generality of our results, and the absence of experimental manipulations of male experience (lines 405-474).

Furthermore, in our Acknowledgement section, we have stated the role the funders took in the study: „The funders had no role in study design, data collection and analysis, decision to publish, or preparation of the manuscript."

We believe that these revisions have significantly strengthened the manuscript and that we have addressed all concerns raised by the reviewers. We have included (i) a detailed response to reviewers, (ii) a tracked-changes version of the manuscript, and (iii) a clean version of our manuscript.

We thank you again for the opportunity to revise our work and we hope that the revised manuscript now meets the journal’s and reviewers’ expectations.

With kind regards,

Ulrike Scherer

(on behalf of all co-authors)

Reviewer #1:

This manuscript presents a study examining paternal effects in the gynogenetic Amazon molly. The research question is intriguing and relevant, and the manuscript is generally well written. However, I have several concerns regarding both methodology and analysis that need to be addressed to strengthen the validity of the findings. The core issue lies in the interpretation of paternity in a system where males do not contribute genetically, raising questions about the assignment of paternal identity. Additionally, there are analytical and reporting elements that require clarification or revision.

# We sincerely thank the reviewer for their thoughtful and constructive feedback. Their careful evaluation and insightful suggestions have substantially strengthened the clarity, precision, and overall quality of our manuscript. We greatly appreciate the time and expertise they invested in this review.

Major Comments:

1. Interpretation of “Paternity” in Gynogenetic Species:

It is problematic to frame the analysis around the assignment (or lack thereof) of paternity to specific males when these males do not contribute genetic material. I recognise that this is probably a simpler way to frame it, to call it sired, fathered, etc. rather than calling it the identify of males that were housed with females, but it’s not technically accurate? Also, as females were exposed to multiple males across time, stored sperm may influence brood outcomes, meaning the most recent male may not be the one affecting brood traits. While the manuscript cites evidence for strong fertilization bias, this comes from a different species (P. reticulata), which is not gynogenetic. It is unclear whether sperm precedence operates similarly in the Amazon molly. The authors should clarify this assumption or provide more robust justification for its applicability here, and address this potential limitation in the discussion.

# Regarding the first point raised (terminology around paternity): We thank the reviewer for this important comment and for drawing attention to the nuanced interpretation of “paternity” in gynogenetic systems. We fully agree that terms such as “sired” or “fathered” can be misleading in a system where males do not contribute genetically to reproduction. To avoid this ambiguity, we are now explicitely acknowledge the paradox of testing for paternal effects in a species without genetic paternity – both in the revised title of our manuscript (lines 2-3) as well as in the Introduction section (lines 92-94). Furthermore, we have now revised the manuscript to phrase this aspect more carefully. Specifically, we are now referring to ‘males that have been assigned to offspring/broods’. For example:

Original manuscript (lines 154-155): “For each brood, the father was identified as the male who was in the female’s tank 30 days before parturition.”

Revised manuscript (lines 211-214): “For each brood, the male who was in the female’s tank 30 days before parturition was identified as the ‘primary male’ […] this male most likely triggered embryonic development of the brood in question.”

Original manuscript (lines 224-226): “However, our results are robust with respect to the removal of offspring or broods, which were sired by males who died within two months post pseudo-fertilization…”.

Revised manuscript (lines 376-377): “…we repeated our main analysis excluding all broods and offspring assigned to males (primary and secondary) that died within two months after mating.”.

Regarding the second point raised (sperm storage): We thank the reviewer for highlighting this important point. We fully agree that our original manuscript did not sufficiently consider the potential role of multiple males’ sperm being present in the female reproductive tract at the time of insemination or during embryonic development. In our revised version, we have now explicitly addressed this issue by identifying not only the male most likely to have triggered embryonic development based on the timing of exposure (the “primary male”) - but also the male whose sperm was second most likely to have influenced embryonic development (“secondary male”) (lines 210-221). We have not identified third and fourth males because our sample size is limited and because we expect more distant encounters with males to be of less importance (lines 219-221). In our statistical analyses, we now test for potential effects of both the primary and secondary males’ identities or body sizes (lines 279-280). In short, we now find significant, yet weak, effects of both males (primary and second) (lines 306-315).

2. Tank Effects:

The manuscript mentions a test for “tank system” but does not appear to include individual tank identity (tank ID) as a factor. This is potentially problematic, as tank-level variation could confound the results. A random effect for tank ID should be included, or at minimum, tank effects should be explicitly tested and reported.

# We thank the reviewer for this important comment and fully agree that tank-level variation can represent a potential confounding factor. In our experiment, each female was housed in a single, uniquely assigned tank throughout the entire study, meaning that female identity and tank identity were synonymous. Accordingly, our statistical models already included this factor as a random effect, called “female identity”. In the revised manuscript, we have now clarified this point in the Methods section of the revised manuscript (lines 151-155 and lines 279-281) and replaced “female identity” with “female/tank identity” wherever applicable (lines 280 and 284, Supporting Information S01 – S10 Table).

In addition, following the reviewer’s suggestion, we have now extended our statistical models to account for potential systematic spatial variation among tanks. Specifically, while we have controlled for variation among tank systems in our original manuscript, we are now also controlling for tank level (tanks were distributed over four vertical levels) and tank centrality (tanks were classified as central, with neighboring tanks on both sides, or peripheral, i.e., edge tanks with only one neighbor) as fixed covariates in all analyses (lines 272-276, Supporting Information S01 – S10 Table). This expanded model structure ensures that any systematic environmental differences across tank positions are appropriately controlled for.

3. Inclusion of Males with Only One Brood:

Figure 1 indicates that some males sired only a single brood. What is the justification for including these males in the repeatability analysis? These single observations may not contribute meaningfully to the estimation of variance components and could dilute the signal. The authors should test whether excluding these cases changes the results and report accordingly. Likewise, there are a lot of factors in the models, many of which are not significant. Have you tried running he models without non-significant fixed effect/covariates? Though I note that it doesn’t seem like it will change the results based on the data plotted in the figures.

# Regarding the first point raised (the inclusion of males with only one brood): All males were included in the repeatability analysis, including those who produced only a single brood, to ensure that the full distribution of reproductive outcomes in the population was accurately represented. To address the reviewer’s concern, we have now clarified this rationale in the Results section and additionally conducted a robustness analysis excluding single-brood males. In short, our results were largely robust to the exclusion of these males, except that the primary male’s repeatability for brood size increased notably (from roughly 2% to 12%; lines 355-371, Supporting Information 3). In our revised manuscript, we clearly highlight the difference in results between our main analysis including all males vs. the analysis without single-brood males and discuss its implications (lines 355-371, lines 418-426).

Regarding the second point raised (removal of non-significant predictors): We have now tested the robustness of our results with respect to the removal of non-significant factors using backward model selection. Our results were largely robust to the exclusion of non-significant fixed effects, except that the weak association between male body size and offspring size reported in our main analyses became non-significant (lines 385-392). In the revised manuscript, we now clearly highlight the differences among these analytical variants and discuss their implications (lines 418-426).

4. Random Effects Structure:

It is unclear why the interaction between male and female identity was not included as a random effect. Given the experimental design, such an interaction could capture potential dyad-specific variation in brood traits?

# We thank the reviewer for this thoughtful suggestion. We agree that, in principle, including a male×female interaction term could capture variation that is specific to a particular pairing. However, in our experimental design, each male was paired with a female only once, resulting in a single brood per unique male–female combination. As there are no repeated measurements within these combinations, the male×female interaction cannot be estimated as a random effect in a meaningful way. For this reason, our models include male ID and female ID as random effects, but not their interaction. In the revised manuscript, we have now added this explanation to the description of model structures (lines 286-289).

5. Experimental Design: It would help if you could state how often females received a new unfamiliar male - it was every two weeks, but for how long, continuously from 70 to 260 days old for some females? And this was just until they produced the 1st brood, yes, and then it was a new male after every new brood? Or did the 2-week cycle start again for this next brood? On line 138 it says that in phase 2, an unfamiliar male was added whenever she gave birth to a new brood, but this was 7 days after giving birth to the new brood, yes?

# We thank the reviewer for this helpful comment, which prompted us to clarify our description of the experimental procedure. In the revised Methods section, we now provide a more detailed explanation of how males were introduced during the two phases of the breeding design and why two different breeding procedures were used (lines 165-189). Specifically, we explain that in Phase 1, beginning at an age of 70 days, each female was continuously paired with an unfamiliar male, who was replaced every two weeks until she produced her first brood. After the first brood, the current male remained for an additional seven days to provide fresh sperm. In Phase 2, we then introduced a new unfamiliar male only when a female gave birth to a new brood, leaving the male in the tank for exactly seven days. While Phase 1 was necessary to effectively initiate reproduction, the transition to Phase 2 minimized potential confounds arising from sperm of multiple males being simultaneously present within a female’s reproductive tract.

To further enhance clarity, we have added a schematic of our experimental procedure (Figure 1) that visually depicts the experimental procedure across both phases, and we now also specify the duration of phase 1 by reporting the mean age at first reproduction for females in the final dataset (N = 54 females, 181.1 ± 39.8 days; lines 175-176).

We hope that these changes make the design and timing of male introductions fully transparent.

Minor Comments / Line-by-Line Suggestions:

• Line 54: Add references to support the statement: “…from insects to mammals.”

# In the revised manuscript, we have now added references for this statement (line50).

• Line 70: Clarify wording. The cited study did not find significant effects of male species; it reported a non-significant trend in the allopatric vs. sympatric comparison.

# In the revised manuscript, we have now rephrases this sentence accordingly (lines 96-100).

• Line 93: Replace “mollies” with “molly.”

# Corrected (line 143).

• Line 118: Replace “juveniles” with “juvenile.”

# Corrected (line 151).

• Line 133-138: Clarify the male exposure schedule. Was an unfamiliar male introduced every two weeks from 70 to 260 days of age, or just until the first brood? After the first brood, were females exposed to a new male each time they reproduced? Line 138 states that the new male was introduced seven days after parturition—this needs to be clearer and more consistent with earlier descriptions.

# In the revised Methods, we now clearly describe how males were introduced in each phase - continuous two-week rotations until first reproduction in Phase 1, followed by seven-day introductions after each brood in Phase 2 (lines 165-182). To further aid clarity, we have added a schematic of the experimental procedure (Figure 1).

For a more detailed answer see also our response to Reviewer 1’s above comment #5.

• Line 148: Remove “a” from “the day they gave a birth.”

# Corrected (line 193).

• Line 178: The sentence “no effect of either factor” is followed by three factors. Please revise for consistency.

# In the revised manuscript, we have now rephrased to “none of these factors significantly affected…” (line 259).

• Line 191: Remove the comma: change “(32), (33)” to “(32,33).”

# Corrected (line 265).

• Line 197: Add “the” before “response.”

# Corrected (line 271).

• Line 211-212: Why is average male length used? If brood-specific male length was measured, it would seem more appropriate and accurate to use that value.

# We used average male length rather than brood-specific male length, as poeciliid ma

---

## [Decision Letter · Decision Letter 1]

17 Dec 2025

PONE-D-25-37087R1Paternal effects without paternity? Testing non-genetic male influence on offspring size and brood size in a gynogenetic vertebrate, the Amazon molly (Poecilia formosa)PLOS One?

Dear Dr. Scherer,

Thank you for submitting your manuscript to PLOS ONE. After careful consideration, we feel that it has merit but does not fully meet PLOS ONE’s publication criteria as it currently stands. Therefore, we invite you to submit a revised version of the manuscript that addresses the remaining points raised during the review process.

We look forward to receiving your revised manuscript.

Kind regards,

Michael Schubert

Academic Editor

PLOS One

Journal Requirements:

Reviewers' comments:

Reviewer's Responses to Questions

**Comments to the Author**

Reviewer #1: (No Response)

Reviewer #2: All comments have been addressed

2. Is the manuscript technically sound, and do the data support the conclusions?

Reviewer #1: Yes

Reviewer #2: (No Response)

3. Has the statistical analysis been performed appropriately and rigorously?

Reviewer #1: Yes

Reviewer #2: (No Response)

4. Have the authors made all data underlying the findings in their manuscript fully available?

Reviewer #1: Yes

Reviewer #2: (No Response)

5. Is the manuscript presented in an intelligible fashion and written in standard English?

Reviewer #1: Yes

Reviewer #2: (No Response)

Reviewer #1: Excellent work revising the manuscript. I have just a few comments for the authors to consider.

Line 34: Just say “this effect was minimal”. Same as at line 311.

Line 125: What does it mean to be second-most likely to be involved? Involved how?

Line 140 and line 213: I would actually argue here that you’re less like to see this sperm precedence in the molly though. The males do not contribute genetically to fertilisation, so there’s been no selective pressure for post-copulatory sperm competition – presumably they would just be a mix in the reproductive tract. On the other hand, I would think there would have been strong selection for sperm longevity within the reproductive tract, to ensure that females have sperm available if they don’t encounter a male.

Line 175: I like the new figure 1, though arguably it could just be in the supplement. I’m trying to full understand the text and figure. The females produced a brood, the male stayed with the female for seven more days. The female was then left alone until she produced a second brood, and then the male was added? So in the figure, male 3 triggered the 2nd brood. So male 4 is the trigger for brood 3. This is confirmed at line 215. So the analyses are all based on broods 2+. If that is not correct, then I’m still missing something. Why is there a brood #1 plotted in figure 2? On that note, why brood number on the x-axis – why not male identity?

Line 222: Why would there not be a male in the tank 30 days prior?

Line 270: Do you mean 128 as noted above at line 234?

Line 391: Check p-value, 0.509? Is it 0.0509?

Figure 2: Given that these are different males – should there be lines joining the points?

In your figures, it might be worth defining primary and secondary males, so that people don't need to read the methods in detail. Something simple like, male the female was housed with and previous male the female was housed with? Something like that?

Reviewer #2: Overall, I found the paper to be much improved and applaud the authors on their careful revisions. I have a few, very minor comments, but am happy with the revisions!

Lines 27-28- I would remove the sample sizes from the abstract, because it feels misleading given that they all descended from one female

Line 79- red, not read

Line 110- expected

Line 381- remove the ‘and’. Generally, there are a number of sentences that start with ‘and’ that are incomplete sentences.

**Do you want your identity to be public for this peer review?** For information about this choice, including consent withdrawal, please see our Privacy Policy

Reviewer #1: No

Reviewer #2: No

---

## [Author Response · Author response to Decision Letter 2]

23 Dec 2025

Reviewer #1

Excellent work revising the manuscript. I have just a few comments for the authors to consider.

# We thank the reviewer for their positive evaluation of our revision.

Line 34: Just say “this effect was minimal”. Same as at line 311.

# Corrected (now lines 34 and line 427).

Line 125: What does it mean to be second-most likely to be involved? Involved how?

# We agree that the original phrasing was insuffiently precise. In the revised manuscript, we have rephrased the sentence to clarify that “second-most likely to be involved” refers specifically to the likelihood that a male’s sperm triggered embryogenesis (now line 125).

Line 140 and line 213: I would actually argue here that you’re less like to see this sperm precedence in the molly though. The males do not contribute genetically to fertilisation, so there’s been no selective pressure for post-copulatory sperm competition – presumably they would just be a mix in the reproductive tract. On the other hand, I would think there would have been strong selection for sperm longevity within the reproductive tract, to ensure that females have sperm available if they don’t encounter a male.

# We thank the reviewer for this thoughtful comment. We agree that, in a gynogenetic system, there is no direct selective pressure on males for post-copulatory sperm competition in the strict sense, given that male genetic material is not incorporated into the offspring. However, we note that Atlantic molly males used in our study evolve primarily within a sexually reproducing system, in which post-copulatory processes are subject to selection. As a result, sperm traits shaped by sperm competition in the sexual context may still be expressed when these males mate with Amazon molly females, even if they do not contribute genetically to reproduction in this particular interaction. At the same time, we agree that female-mediated post-copulatory processes may be dampened in the Amazon molly compared to sexually reproducing species. We have therefore revised the manuscript to explicitly note that it remains unclear whether such fresh-sperm precedence operates to the same extent in the Amazon molly (now lines 141-142).

We further agree with the reviewer that selection for sperm longevity is likely to be important in this system, ensuring that females have access to sperm even in the absence of male encounters. We believe that sperm longevity is already accounted for in our analyses through the explicit consideration of both primary and secondary males.

Line 175: I like the new figure 1, though arguably it could just be in the supplement. I’m trying to full understand the text and figure. The females produced a brood, the male stayed with the female for seven more days. The female was then left alone until she produced a second brood, and then the male was added? So in the figure, male 3 triggered the 2nd brood. So male 4 is the trigger for brood 3. This is confirmed at line 215. So the analyses are all based on broods 2+. If that is not correct, then I’m still missing something. Why is there a brood #1 plotted in figure 2? On that note, why brood number on the x-axis – why not male identity?

# We agree with the reviewer that the new Fig. 1 could be a supplementary figure. However, we believe that understanding the experimental procedure - particularly the definition of primary and secondary males - is essential for understanding the results, and this information is most accessible when presented visually (as not all readers will read the detailed Methods section). Therefore, keeping it in the main text adds substantial value.

The reviewer’s interpretation of the breeding protocol is entirely correct: after the first brood, the male remained with the female for seven additional days, after which the female was housed alone until the next brood, at which point a new male was introduced; and our main analyses are indeed based exclusively on brood numbers ≥2. To make this clearer, we have revised both the Methods section (lines 169-171, 176-183) and Fig. 1 description (lines 192-196). In addition, we have updated Fig. 1 to visually emphasize that the first brood is not included in the main analyses.

Regarding the illustration of Brood 1, we agree that without explicit clarification this is misleading, as Phase 1 reproduction is excluded from the main analyses. To address this, we have now highlighted Brood 1 in a different color and added a note to the figure caption, stating that this brood is shown for completeness only and is not included in our main analyses (lines 195-196).

Finally, with respect to the x-axis in Fig. 2, we agree that an alternative visualization using male identity on the x-axis would be possible. However, we believe that the current representation conveys the repeated-measures structure of the data and the underlying experimental logic equally well.

Line 222: Why would there not be a male in the tank 30 days prior?

# In these cases, no male was present in the female’s tank 30 days prior to parturition because we did not record a brood at that time. We have revised the sentence to clarity this point (lines 230-231).

Line 270: Do you mean 128 as noted above at line 234?

# We thank the reviewer for catching this inconsistency. The correct sample size is 127 broods (not 128, as stated in the previous line 234). In the revised manuscript, we have corrected this (now line 243).

Line 391: Check p-value, 0.509? Is it 0.0509?

# We thank the reviewer for carefully checking this value. The reported p-value of 0.509 is correct and not a typo. We detected a trend for the secondary male only. The original wording was misleading in this regard. We have now revised the sentence to accordingly (now line 403-404).

Figure 2: Given that these are different males – should there be lines joining the points?

# In Fig. 2, points corresponding to broods from the same male are indeed connected by lines to indicate repeated contributions by the same male across broods. We suspect that this may not have been clearly visible in the previously uploaded version of the figure, potentially due to file resolution or formatting issues during submission. We have therefore uploaded a new, higher-quality version of Figure 2 and hope that this resolves the issue and makes the connecting lines clearly visible.

In your figures, it might be worth defining primary and secondary males, so that people don't need to read the methods in detail. Something simple like, male the female was housed with and previous male the female was housed with? Something like that?

# We thank the reviewer for this helpful suggestion. We agree that defining primary and secondary males directly in the figure caption improves clarity. We have now added such a brief explanatory note to the figure caption (lines 337-340, 350-354).

We would like to thank the reviewer once again for their thoughtful and constructive feedback. Their careful reading and insightful comments were instrumental in improving the overall quality of the manuscript, and we are very grateful for the time and effort they invested in this review.

Reviewer #2

Overall, I found the paper to be much improved and applaud the authors on their careful revisions. I have a few, very minor comments, but am happy with the revisions!

# We thank the reviewer for their positive evaluation of our revision.

Lines 27-28- I would remove the sample sizes from the abstract, because it feels misleading given that they all descended from one female

# We agree that clarity about the biological origin of the study animals is essential, and we have therefore now added the information that all Amazon molly females derive from a single clonal lineage to the abstract (line 28). We would, however, prefer to retain the sample sizes in the abstract. The numbers are intended to convey the scale and experimental effort of the breeding design - particularly the large number of broods and offspring generated and the extensive involvement of different males - rather than to imply independent female genetic backgrounds. We therefore feel that, with the added clarification, the sample sizes remain informative and appropriate to include in the abstract.

Line 79- red, not read

# Corrected (now line 79).

Line 110- expected

# Corrected (now line 110).

Line 381- remove the ‘and’. Generally, there are a number of sentences that start with ‘and’ that are incomplete sentences.

# In the revised manuscript, we have corrected the incomplete sentence in line 381 (now line 394). Additionally, we have corrected all other incidences with the same pattern (now lines 78, 426, 461).

We would like to thank the reviewer once again for their thoughtful and constructive feedback. Their careful reading and insightful comments were instrumental in improving the overall quality of the manuscript, and we are very grateful for the time and effort they invested in this review.

---

## [Editor Report · Decision Letter 2]

5 Jan 2026

Paternal effects without paternity? Testing non-genetic male influence on offspring size and brood size in a gynogenetic vertebrate, the Amazon molly (Poecilia formosa)

PONE-D-25-37087R2

Dear Dr. Scherer,

We’re pleased to inform you that your manuscript has been judged scientifically suitable for publication and will be formally accepted for publication once it meets all outstanding technical requirements.

Kind regards,

Michael Schubert

Academic Editor

PLOS One

---

## [Editor Report · Acceptance letter]

PONE-D-25-37087R2

PLOS One

Dear Dr. Scherer,

I'm pleased to inform you that your manuscript has been deemed suitable for publication in PLOS One. Congratulations! Your manuscript is now being handed over to our production team.

Kind regards,

on behalf of

Dr. Michael Schubert

Academic Editor

PLOS One